



# Carbon accumulation rates of Holocene peatlands in central-eastern Europe document the driving role of human impact for the past 4000 years

Jack Longman[1], Daniel Veres[2], Aritina Haliuc[2,3], Walter Finsinger[4], Vasile Ersek[5], Daniela Pascal[6,7], Tiberiu Sava[6,7], Robert Begy[8]

[1]Marine Isotope Geochemistry, Institute for Chemistry and Biology of the Marine Environment (ICBM), University of Oldenburg, 26129, Oldenburg, Germany
[2] Romanian Academy, Institute of Speleology, 400006 Cluj-Napoca, Romania
[3]EPOC, UMR 5805, Université de Bordeaux, Pessac, France
[4]ISEM, Univ Montpellier, CNRS, EPHE, IRD, 34095, Montpellier Cedex 5, France
[5]Department of Geography and Environmental Sciences, Northumbria University, Newcastle-upon-Tyne, NE1 8ST, UK
[6]RoAMS Laboratory, Horia Hulubei National Institute for Physics and Nuclear Engineering, Reactorului 30, 077125, Măgurele-Bucharest, Romania
[7]University of Bucharest, Faculty of Geography, Bd. Nicolae Bălcescu 1, 030018, Bucharest, Romania
[8]Interdisciplinary Research Institute on Bio-Nano-Science, Babes-Bolyai University, Treboniu Laurian 42, 400271, Cluj-Napoca, Romania

*Correspondence to*: Jack Longman (jack.longman@uni-oldenburg.de)

**Abstract.** Peatlands are one of the largest terrestrial carbon sinks on the planet, yet little is known about carbon accumulation rates (CARs) of mountainous examples. The long-term variability in the size of the associated carbon sink and its drivers remain largely unconstrained, especially when long-term anthropogenic impact is also considered. Here we present a composite CAR record of nine peatlands from central-eastern Europe (Romania and Serbia) detailing variability in rates of carbon accumulation across the Holocene. We show examples of extremely high long-term rates of carbon accumulation (LORCA >120 g C m$^{-2}$ yr$^{-1}$), indicating that at times, mountain peatlands constitute an efficient regional carbon sink. By comparing our data to modelled palaeoclimatic indices and to measures of anthropogenic impact we disentangle the drivers of peat carbon accumulation in the area. Variability in early and mid-Holocene CARs is linked to hydroclimatic controls, with high CARs occurring during the early Holocene and lower CARs associated with the transition to cooler and moister mid-Holocene conditions. By contrast, after 4000 years (calibrated) before present (yr BP) the trends in CARs indicate a divergence from hydroclimate proxies, indicating that other processes became the dominant drivers of peat CARs. We suggest that enhanced erosion following tree cover reduction as well as enhanced rates of long-distance atmospheric dust fallout might have played a role as both processes would result in enhanced mineral and nutrient supply to bog surfaces, stimulating peat land productivity. Surprisingly though, for the last 1000 years, reconstructed temperature is significantly correlated with CARs, with rising temperatures linked to higher CARs. We suggest under future climate conditions, predicted to be warmer in the region, peat growth may expand, but that this is entirely dependent upon the scale of human impact directly affecting the sensitive hydrological budget of these peatlands.



## 1 Introduction

Peatlands are some of the most proficient mediums for long-term carbon (C) sequestration in the terrestrial environment (Loisel et al., 2014), with the C sink in globally distributed peatlands estimated to be as high as c.600 gigatonnes (Gt) C (Dargie et al., 2017; Yu et al., 2010). This carbon sink has developed primarily during the Holocene, but rates of

development vary both temporally and spatially, driven by shifts in nutrient availability (Kylander et al., 2018; Ratcliffe et al., 2020), moisture and temperature (Charman et al., 2013; Loisel et al., 2014; Yu et al., 2010), and the speed at which accumulated peat decomposes (Clymo et al., 1998). Specifically, warmer and wetter climates tend to lead to higher peat carbon accumulation rates (CARs) (Loisel et al., 2014), as warmer temperatures stimulate peat growth that offsets organic matter decomposition (Charman et al., 2015, 2013; Yu et al., 2010).

As such, understanding the history of carbon accumulation in peatlands is important for critically assessing the terrestrial carbon cycle in our anthropogenically altered world (Hugelius et al., 2020; Loisel et al., 2021). Model-derived estimates of future climatic change suggest that peatlands will act as an efficient carbon sink as climate warms (Gallego-Sala et al., 2018), as observed in reconstructions of warm periods in Earth's past (Treat et al., 2019). However, this conclusion is far from certain, as land-use driven degradation of peat since the 1960s have led to peatlands acting as a net source of carbon

(Leifeld et al., 2019). Land use and peatland degradation played a major role in anthropogenic carbon release long before the 1960s, with evidence that humans progressively altered the terrestrial carbon cycle for the last 7000 years (Kaplan et al., 2011; Ruddiman et al., 2011). Greater land exploitation leads to enhanced carbon release, so this trend will likely continue as human pressure on land use is expected to increase (Kaplan et al., 2011; Ruddiman and Ellis, 2009).

On individual scales, fire and grazing disturbances negatively impact peatland development (Nieveen et al., 2005; Worrall

and Clay, 2012), due to the loss of surface peat or changes to local vegetation (Garnett et al., 2000; Kuhry, 1994). Furthermore, draining of wetlands and the extraction of peat plays a major role in both long-term carbon accumulation in peat and its short-term release back into the atmosphere (Holden et al., 2011; Turetsky et al., 2011), with hydrological shifts potentially altering the ecology of bogs (Tahvanainen, 2011). Furthermore, climate change also influences the stability of the extensive northern hemisphere boreal peatlands, potentially resulting in the enhanced release of methane, thus further

exacerbating the effects of rising global temperatures (Schaefer et al., 2014; Schuur et al., 2015). Finally, it is possible that increased erosion as a result of forest removal either by human actions or natural causes may lead to increased mineral in-wash, stimulating peat growth (Kylander et al., 2018; Longman et al., 2017a). Due to the complex relations between these factors, and the equally complex response of peat environments, the interaction of anthropogenic disturbances and peat growth over long timescales, and on peat CARs is as yet unclear.

Mountain bogs are rarely considered in estimates of carbon storage (Chen et al., 2014), likely due to their limited extent when compared to tropical and boreal marshes and peatlands (Dargie et al., 2017; Loisel et al., 2014). This lack of studies extends to the Carpathian Mountains, the second largest European mountain range, which stretches for nearly 1500 km from Czechia to Serbia (Fig. 1). In the Carpathians, only one high-resolution study of CARs has been completed at Tăul Muced,



Romania, indicating that about 250 tonnes (t) C may have been sequestered over the past 7000 years (Panait et al., 2017). This is insignificant in isolation, but when all mountain peatlands from this mountain range (Pop, 1960), which include substantially larger bogs such as Mohos in eastern Romania, roughly 40 times larger (Longman et al., 2017b), are considered, it may be that Carpathian bogs constitute an important carbon sink for the overall carbon budget.

Here we present CARs for nine bogs in central-eastern Europe, reconstructing variability of carbon accumulation for the past 10,000 years. By combining these reconstructions with changepoint-modelling and proxies of past hydroclimatic variability, we assess the drivers of carbon accumulation in mountain regions. We provide a first estimate of carbon storage in these environments, and investigate the extent to which human activity in mountain regions may have impacted on the peatland accumulation rates throughout the Holocene.

## 2 Methods and Materials

### 2.1 Site Locations and Coring

We analysed the organic matter content of 8 bogs in Romania (n=6) and Serbia (n=2) (see Table 1), and combined it with data from the previously published CAR from Tăul Muced (TM) bog (Panait et al., 2017). All new sites in this study were cored using a Russian corer, with samples immediately wrapped in cling film and shipped to laboratories for description and analysis.

### 2.2 Age Modelling

For all bogs, radiocarbon accelerator mass spectrometry (AMS 14C) dating analyses were completed to estimate the age of peat accumulation. The samples from Despotovac (DES), Mluha (MLH) and Iezerul Mare (IZM) sites were pre-treated and measured at RoAMS laboratory in IFIN-HH according to the protocols described in Sava et al. (2019). A summary of the radiocarbon dates and laboratory procedures for each investigated bog is found in Supplementary Table 1. Age-depth modelling was completed using Bacon (Blaauw and Christen, 2011), with ages calibrated using IntCal20 (Reimer et al., 2020) and the models presented in Supplementary Figures 1-8. For TM, we used the published age model of Panait et al. (2017).

### 2.3 Carbon Accumulation Rate (CARs) Calculations

For each bog samples of exactly 1cm3 were taken and dried before being weighed to estimate dry bulk density. Organic content was then measured via loss-on-ignition, performed in a muffle furnace at 550°C, and calculated by comparing the weight pre- and post-ignition (Heiri et al., 2001; Veres, 2002). Samples from Sureanu (SUR; Longman et al., 2017a) and Mohos (MOH; Longman et al., 2017b) were analysed at the University of Northumbria. Samples from Baia Vulturilor (BVU), Zanoaga Rosie (ZNG), Despotovac (DES), Mluha (MLH) and Iezerul Mare (IZM) were analysed at the Institute for Speleology, Romanian Academy in Cluj-Napoca, whilst samples from Crveni Potok (CP; Finsinger et al., 2017) were



analysed at the University of Montpellier. Carbon accumulation rates (CAR) were calculated by dividing the carbon content

of each sample by the modelled peat accumulation rate following Ratcliffe et al. (2020) (Fig. 2).

### 2.4 Statistical Analysis

To estimate the size of the carbon sink represented by the studied bogs, we calculate long-term apparent carbon accumulation rates (LORCA) (Clymo et al., 1998; Turunen et al., 2002) and the amount of carbon stocked by the bogs. LORCA is calculated by multiplying the median carbon (C) content by the median density and the depth of the peat

accumulation, before dividing through by the basal age of the bog. This provides a first order estimate of the rate at which each bog accumulated carbon on a long-term scale, and can be used to calculate the carbon stock (in gigatonnes, Gt), represented by each bog using the following equation:

$$C\ stock\ (Gt) = (A \times h) \times \overline{\rho} \times \overline{c}/10^9 \qquad\qquad\qquad \text{(Eq. 1)}$$

Where $A$ is the peatland area (in km$^2$), $h$ is the peat depth (in cm), $\overline{\rho}$ is the median peat density (in g cm$^{-3}$) and $\overline{c}$ is the median

carbon content (in %) (Page et al., 2011).

To investigate the time series data generated by this approach, we use changepoint-modelling (Gallagher et al., 2011). This allows for the inference of statistically significant shifts in the dataset, meaning we can better investigate the drivers of change. To investigate the impact of past climate and moisture variability on the CARs of individual sites in discreet periods, we calculate average CARs for three periods of both climatic and human-impact interest: the Roman Warm Period (RWP;

2200 – 1550 yr BP), the Medieval Warm Period (MWP; 950 – 1250 CE) and the Little Ice Age (LIA; 1300 – 1850 CE). We also calculated the apparent recent rate of carbon accumulation (RERCA) for the period from 50 yr BP to present (Table 1). However, the RERCA values should be treated with caution because CAR values in the uppermost layers of peat may vary due to the exponential nature of peat decay (Clymo et al., 1998; Young et al., 2019). Therefore, we also quantified the amount of carbon accumulation for the past 1000 years (see Table 2 for all average values), following Gallego-Sala et al.

120    (2018).

To estimate the total size of peatland carbon sink in the Romanian Carpathians selected due to the availability of data survey on peatland extent, we used the mean carbon content and the mean bulk density from the 8 peat bogs, and total bog area and bog depth from Pop (1960), and applied Equation 1 to these data. To account for uncertainties, we used a 1-SD for our new data, and applied a 10% error to Pop (1960) prior to estimating the peatland carbon sink with Monte Carlo simulations

(Supplementary Table 2). Repeat Monte Carlo simulations (n=10,000) were employed to estimate each of the variables, using the R package truncnorm, and boundaries dictated by the data.

### 2.5 Composite CAR record and data-model comparison

To interpret the effect of past hydroclimatic drivers on the Romanian Carpathian mountain peatlands, we synthesise all datasets, resulting in a representative record of CAR across the Middle and Late Holocene in the region (Fig. 3). To achieve

this, for each individual site, we bin all data at 50-year intervals. For these bins, we consider the 25 years either side of the





bin age (i.e., the 0 yr BP bin covers the period 26 to -25 yr BP). To allow for comparison across sites, we convert these raw binned CAR data into z-scores. We then calculate mean and standard deviation values for each of the bins, for all studied sites. This approach allows for the interpretation of large-scale changes in CAR for a region across a long time period. As individual CARs are highly variable, with numerous stochastic and site-specific forcing factors, averaging multiple records
provide insights into broad scale factors controlling regional CARs (Loisel et al., 2021; Marlon et al., 2016).

To compare our composite CAR output to modelled palaeoclimatic indices, we use data from PaleoView (Fordham et al., 2017). From PaleoView, we extracted 6 parameters related to past precipitation and temperature variability, with each temperature variable expressed as a z-score relative to recent (i.e., present day set as 1975 CE, or -25 yr BP in the model). For each variable, we use the mean area extracted from the region 40° – 47.5 °N, 17.5° – 27.5 °E, which covers all sites in
this study (Fig. 1).

To investigate the effect of other potential drivers, we compare the composite output to three measures of human impact and land use (population, area of pasture and area of cropland) derived from the HYDE dataset (Goldewijk et al., 2017). Due to the nature of the dataset, we use the "Central Europe" region, as it contains both Romania and Serbia (Goldewijk et al., 2017). We also explore an estimate of the intensity of land use in the region, from Kaplan et al., (2011), which represents a
proportion of the 5 degree grid square (centred on 46 °N, 24 °E) utilised for any type of human activity on the landscape. To investigate the impact of dust deposition, we used two regionally representative reconstructions of dust deposition through the Holocene, from MOH (Longman et al., 2017b), and TM (Panait et al., 2019). To investigate the impact of biomass burning on the record, we compare with the compilation of eastern European charcoal deposition from Feurdean et al. (2020), from which we extract the boreal (BOR) record, as this covers all sites in this study. We used linear interpolation to
bring all datasets onto a 50-year timescale for comparison with our binned composite CAR record.

## 3 Results

### 3.1 Basic peat bog information

Where possible, the basal peat layer was reached, allowing for a reliable estimate of the total C stock. This varied from 940 cm at MOH to 193 cm at BVU, reflected in variable accumulation rates among records (Table 1). For example, ZNG in the
Semenic Mountains, containing a long hiatus or very low accumulation rates in peat accumulation, yielded accumulation rates (ARs) as low as 0.02 cm yr-1. In contrast, IZM and DES displayed ARs of greater than 0.1 cm yr-1, corresponding to c. 800 cm of peat deposition in c. 6200 years at IZM and c. 200 cm in ~2000 years at DES (Table 1). The lowest average density is 0.07 g cm-3 at MOH, reflecting the pure Sphagnum composition of the peat stock. In contrast, the highest density is observed at DES (0.35 g cm-3), where minerogenic material has been delivered to the mire from the surrounding slopes
impacted by agriculture. In terms of median carbon content, all records show values above 50 % C other than DES, where minerogenic input leads to a C content of c. 20% (Table 1). The highest median C values are recorded at ZNG, where C represents 57.7 % of the peat content.





## 3.2 LORCA, period CARs and estimates of total carbon storage

Calculations of LORCA indicate a wide range of long-term accumulation of carbon between records. The lowest LORCA
values are from ZNG (17 g C m$^{-2}$ yr$^{-1}$), and at CP, which is a slowly accumulating bog (21 g C m$^{-2}$ yr$^{-1}$). In contrast, the
highest LORCA values are reconstructed from the mountain bogs MLH and IZM, with values of 92 g C m$^{-2}$ yr$^{-1}$ and 121 g C
m$^{-2}$ yr$^{-1}$, respectively (Table 2). When combined with estimates of the thickness and spatial extent of the bogs, we can
calculate total carbon storage in each location. Of our studied locations, MOH contains the most carbon, with an estimate of
0.0003 Pg C, whilst the remainder of the sites contain between 0.00005 and 0.0001 Pg C (Table 2).
CARs for the Roman Warm Period (RWP) vary from 7.6 g C m$^{-2}$ yr$^{-1}$ at DES to 74.03 g C m$^{-2}$ yr$^{-1}$ at MLH, with the majority
of sites between 15 – 25 g C m$^{-2}$ yr$^{-1}$ (Table 2). For the Medieval Warm Period (MWP), the lowest average CAR is 20.44 g C
m$^{-2}$ yr$^{-1}$ at BVU, with MLH displaying the greatest CAR (191.24 g C m$^{-2}$ yr$^{-1}$). For the LIA, values range from 20.85 g C m$^{-2}$
yr$^{-1}$ (CP) to 235.78 g C m$^{-2}$ yr$^{-1}$ (MLH). For the period 1000 – 100 yr BP, all values other than MLH are between 25 and 75 g
C m$^{-2}$ yr$^{-1}$. The RERCA calculations indicate CARs between 15 – 50 g C m$^{-2}$ yr$^{-1}$ for most sites, but with IZM (120.59 g C m$^{-2}$
yr$^{-1}$) and MLH (91.74 g C m$^{-2}$ yr$^{-1}$) well above that range (Table 2).

## 3.3 Carbon Accumulation Rates throughout the Holocene

We developed for each site a time series displaying how CARs varied through the Holocene and combined them into a
composite time series (Fig. 3), which shows changes in CAR at broader spatial scales. For the period 7000 – 4000 yr BP, z-
scores show a slow decrease until a low point of -1 by 4000 yr BP. This is followed by rapid and sustained increases in
CARs, reaching a value of 1 by 2100 yr BP, primarily driven by the large rise in CAR at SUR during this time (Fig. 3). This
is followed by a rapid drop to a period of low values between 2000 – 1300 yr BP, before another sustained period of
increasing CARs between 1500 yr BP and present (Fig. 3).

## 4 Discussion

### 4.1 Long term carbon accumulation, and an estimate of the Carpathian peatland carbon sink

Our data indicate that CAR rates vary greatly both spatially and temporally in the Carpathian Mountains (Fig. 2, 3). Our
LORCA estimates also reflect this trend, with slow-growing peatbogs such as CP accumulating peat at an average of 21 g C
m$^{-2}$ yr$^{-1}$, whilst fast accumulating records such as IZM reach LORCA values as high as 121 g C m$^{-2}$ yr$^{-1}$ (Table 2). IZM and
MLH display some of the highest LORCA values reported in the literature (Loisel et al., 2014), and indicate that raised
ombrotrophic mountain bog environments may be extremely effective carbon sinks. This is supported by comparable
LORCA from bogs in the Andes, where they may reach 250 g C m$^{-2}$ yr$^{-1}$ (Chimner and Karberg, 2008). Typical lowland bogs
sequester carbon at rates of below 30 g C m$^{-2}$ yr$^{-1}$ (Loisel et al., 2014), and so rates of over 100 g C m$^{-2}$ yr$^{-1}$ are notably high.
Indeed, such high CARs are similar to measurements of recent peat accumulation rates which have not been corrected for





peat decomposition (Young et al., 2019). It is possible the location of some mountain bogs at the base of scree slopes, means that some could be well-supplied with regards to nutrients via minerogenic input, such as for SUR (Longman et al., 2019,

2017a). This is analogous to nutrients supplied from volcanic ash deposition, which has been observed to stimulate greater peat accumulation in bogs in New Zealand (Ratcliffe et al., 2020) and Japan (Hughes et al., 2013). In other locations, such as the true raised bogs MLH, IZM and MOH, or blanket bogs BVU and ZNG, a more likely explanation is that the frequency of long-term atmospheric dust deposition in the Carpathian region (Longman et al., 2017b; Panait et al., 2019; Varga et al., 2016, 2013) helps to supply nutrients, a process known to stimulate considerable peat growth (Kylander et al., 2018).

We use our measurements of carbon content and peat density in combination with published estimates of the extent of Romanian Carpathian peatlands to estimate the size of this regional carbon sink (Pop, 1960) (Supplementary Table 2). In Pop (1960), Carpathian peatlands were separated into two main types; oligotrophic (primarily raised bogs, covering c. 15 $km^2$, with an average depth of ~2 m) and eutrophic (including fens and marshes, covering c. 60 $km^2$, with an average depth of 1 m). It must be noted this estimate only includes bogs located in the Carpathian Mountains, and does not include the

extensive lowland wetlands of areas like the Danube valley and its delta (Pop, 1960).

Our modelling suggests oligotrophic bogs, the class in which most mountain bogs fall, represent a mean carbon sink of 0.002 Gt C in the Romanian Carpathians, with the uppermost models (95[th] percentile) indicating 0.01 Gt C is stored. Eutrophic bogs represent a mean carbon sink of 0.004 Gt C, with the 95[th] percentile at 0.008 Gt C. Combined, the peatland sink of the Romanian Carpathians most likely comprises a value of 0.006 Gt C, reaching 0.012 Gt in the 95[th] percentile of models (Fig.

5). In isolation, these estimates of mountain bogs comprise a small proportion (c. 0.002%) of the global peatland carbon sink (roughly 600 Gt) with individual tropical wetland areas comprising orders of magnitude greater carbon stores (Dargie et al., 2017; Warren et al., 2017). However, the rapid nature of carbon accumulation in the Carpathian bogs studied here, and in other mountainous regions such as Tibet (Chen et al., 2014), suggests they may be more efficient sinks than tropical bogs, which accumulate more slowly and decompose faster (e.g., Dargie et al., 2017; Hapsari et al., 2017).

**4.2 Changing rates of carbon accumulation during the Holocene**

By investigating the changepoint models of each individual peatland CAR, and the composite CAR record, we explore changing rates of accumulation through the Holocene (Figs. 2, 3). In the studied region, three peatlands (CP, MOH and ZNG) were established in the early Holocene (10,000 – 7000 BP). For this period, average CARs are above the LORCA of each bog (see Fig. 2), indicating rapid accumulation. This is in agreement with other studies of European peatlands (Ratcliffe

et al., 2018), where rapid CARs are reconstructed in the early Holocene as a result of warm temperatures, a reflection of the key role temperature plays in CARs (Charman et al., 2009). Regionally, our data reflect the impact of warmer than present conditions and lower than present precipitation levels and higher seasonality reconstructed from multi-proxy records (Buczkó et al., 2013; Davis et al., 2003; Feurdean et al., 2008; Magyari et al., 2009; Tóth et al., 2018, 2015) and modelled data which coincided with higher than present (summer) insolation (Berger and Loutre, 1991).





During the mid-Holocene (7000 – 3000 yr BP), CARs were generally low, with our composite record reconstructing a decreasing trend until 4000 yr BP (Fig. 3) and supported by a series of negative changepoints in individual records (Figs. 2,3). CAR rates for this period are below the LORCA for all sites (Fig. 2), indicating a period of slow peat growth on a regional scale. This may be linked to colder and wetter than today climatic conditions which progressively established in the region (Davis et al., 2003; Feurdean et al., 2008; Perşoiu et al., 2017), following a peak in temperatures prior to 8500 yr BP

(Tóth et al., 2015). Cooler conditions and higher lake levels are reconstructed from chironomid records from different parts of the southern Carpathians (Tóth et al., 2015) while pronounced depositional changes and increased lake levels are observed in both the eastern and western Romanian Carpathians (Haliuc et al., 2017; Magyari et al., 2009), and fluvial records indicate greater river flows (Persoiu, 2010). Around 5000 yr BP, a major shift in atmospheric circulation towards a variable NAO-index is shown in water isotope data from Scarişoara cave (Romania; Perşoiu et al., 2017), reflecting continental-scale

changes (Olsen et al., 2012). This coincides with changes in insolation, with strong contrasts between December and June insolation (Berger and Loutre, 1991).

At ZNG, peat accumulation ceased almost entirely for this period (Fig. 2), and BVU started accumulating peat only after 4000 yr BP reflecting the extent to which cold environmental conditions impacted on CARs in the Southern Carpathians. The local nature of CAR responses, however, may indicate local climatic variability, with peat CARs stable at higher

elevation (SUR), and even rising at slightly lower elevation (MLH) during this time (Fig. 2). In our composite record, the lowest levels of carbon accumulation are reflected in the period 4200 – 3800 BP, which coincides with the 4.2 kyr event (Fig. 4). This was a roughly 100-year long period of Europe-wide dry and cool climate (Bini et al., 2019), reflected in the Carpathians by increased dust deposition (Longman et al., 2017b), and evidence for cold and dry (winter) conditions (Constantin et al., 2007; Drăguşin et al., 2014; Perşoiu et al., 2019; Tóth et al., 2015), which appear to have led to region-

wide decreases in peat growth (Figs. 2, 3). However, short-term local increases in precipitation are reconstructed in a mid-latitude lake record (Haliuc et al., 2017), and low-altitude fluvial records (Howard et al., 2004). These conclusions are supported by other parameters such as chironomid-based summer temperatures and testate-amoeba derived measures of wetness (Onac et al., 2002; Schnitchen et al., 2006; Tóth et al., 2015). These contrasting trends reflect seasonal responses to changing local moisture sources, not only in the Carpathians but also throughout the Balkans (Haliuc et al., 2017; Perşoiu et

al., 2019), indicating the particularities and dynamics of the regional climatic variability in southeastern Europe (see Haliuc et al., 2017; 2020).

After this period of low carbon accumulation, CARs rebound, with widespread increases observed, and a series of positive changepoints recorded, notably at MOH, SUR, IZM and CP from 3000 BP onwards (Fig. 2). This is reflected by a rapid increase in our composite of CAR measurements, where a long-term CAR increase is reconstructed between 3400 – 2000 BP

(Fig. 4). It is possible this is linked to higher variability of the climatic conditions during this period, as evidence suggests the period 3500 – 2000 BP was generally cool and wet across the region (Magyari et al., 2009; Schnitchen et al., 2006; Tóth et al., 2015). The specific atmospheric circulation regime of the Carpathians (Haliuc et al., 2017; Longman et al., 2017a), which drives significant lateral hydrological variability across the wider region (Longman et al., 2019; Magyari et al., 2013),





may explain why some pollen-based climatic indices suggest a warming between 3000 – 2400 BP (Feurdean et al., 2008),
and carbon isotopes in peat indicating high climatic variability (Cristea et al., 2013). However, instead of considering only a
climatic control on CARs over this period, it is possible the increasing impact of human activity on the region (Longman et
al., 2017a) contributed to the contradictory proxy reconstructions of climate at this time, with vegetation-based climate
reconstructions potentially impacted by anthropogenic disturbances (Chevalier et al., 2020; Finsinger et al., 2010).

A steady increase in land-use in the Carpathian area has been reconstructed for the last 7000 years using modelled
deforestation rates (Giosan et al., 2012). Since 3000 yr BP increasing land-use has been demonstrated, tracking demographic
variability (Kaplan et al., 2011). Enhanced land use and tree felling in the mid-to-high mountain environments (Hughes and
Thirgood, 1982; Longman et al., 2017a) would have increased erosion (Longman et al., 2019), with increased erosion
observed in lake records from the Balkans at this time (Wagner et al., 2012), and the onset of minerogenic input to SUR bog
(Longman et al., 2017a). Greater erosion would result in increased nutrient supply, in turn stimulating peat growth in a
similar manner to the increased deposition of mineral dust (Kylander et al., 2018), or volcanic ash (Ratcliffe et al., 2020).
This hypothesis is supported by the decreasing CARs reconstructed after 2000 BP during the Migration period denoting
lower levels of land use and less anthropogenic disturbance of the environment (Kaplan et al., 2011; Longman et al., 2018),.
The period of low CARs (2000 – 1300 yr BP) is followed by a series of positive changepoints and a gradual rise in the
composite CAR record until the present day (Figs. 2, 3). The climate during this interval was characterised by a warming
trend in both summer and winter temperatures but with regionally specific hydroclimate patterns (Perşoiu and Perşoiu,
2019). The lack of any relationship between CARs and the apparent warmth of the RWP (2200 – 1500 yr BP) and MWP
(1000 – 700 yr BP) and the cool LIA (650 – 100 yr BP) in the region further support the theory that peat accumulation rates
were impacted by human activity (Fig. 3, Table 2). Indeed, in 6 of our 9 sites, the LIA displays higher CARs than during the
MWP, despite the apparent controlling nature of temperature change on peat accumulation (Charman et al., 2015, 2013).
This may be a feature of the low latitude nature of our sites where, unlike Arctic sites where low temperatures and high
moisture lead to lower CARs, lower temperatures and high moisture stimulate peat growth. However, it is also possible the
lack of any correlation is due to the variable expression of these transient periods of climatic change (MWP and LIA) on the
climate of central-eastern Europe (Roberts et al., 2012). There is evidence which suggests climate in southeastern Europe in
the last 1000 years did not mirror that of western Europe, with a lack of correlation between tree ring derived summer
temperatures in the eastern Carpathians (Popa and Kern, 2009) and central European records (Buntgen et al., 2011). This is
further supported by the disconnection between southeastern European pollen reconstructions of temperature and those from
central and western Europe in the Holocene (Davis et al., 2003; Mauri et al., 2015). But, other work suggests the MWP was
indeed warm and dry in Romania (Feurdean et al., 2011), whilst the LIA was overall cooler (Cleary et al., 2018), with
regional expressions of hydroclimate variability. For the MWP, contrasting hydroclimate conditions are reconstructed from
records across Romania. Warm and dry conditions are evident in records from the north and northwest Romanian
Carpathians (Cleary et al., 2018; Cristea et al., 2013; Forray et al., 2015; Popa and Kern, 2009) whilst records in the north,
south and east Carpathians show wet conditions (Diaconu et al., 2017; Feurdean et al., 2015; Gałka et al., 2016; Panait et al.,



2017, 2019). These apparently contrasting results further indicate the complexity of climate in the region, located as it is at the confluence of three major atmospheric systems (Longman et al., 2017b; Obreht et al., 2016). As such, the range of CARs may be the result of still unexplored local hydroclimatic gradients. As a result, unpicking exactly which features of a climatic record in central-eastern Europe may be attributed to climatic forces, and which may be human impact-related remains a challenge.

## 4.1 Data-model comparison

One way in which potential drivers of change in peat accumulation may be elucidated is by comparing our data to model reconstructions of climatic variables such as temperature and precipitation, and investigating their relationship with the observed shifts in CAR (Fig. 4). Here we compare the results of our composite record (binned at 50-year resolution) to a number of climatic indices extracted at 50-year timesteps from modelling reconstructions of the last 5000 years (Fordham et al., 2017).

When comparing the full datasets, several statistically significant correlations are revealed (Table 3, Fig. 4). These include both temperature and precipitation seasonality, and diurnal and annual temperature range (Table 3). Strong negative correlations between both precipitation and temperature seasonality and CARs indicate that as seasonality increases, CARs decrease. This is in direct contrast to high latitude bogs from Alaska, where it has been shown that increased seasonality during the middle Holocene led to higher CARs (Jones and Yu, 2010). The anticorrelation we observed here suggests that in more temperate bog locations, seasonality, and implied higher summer temperatures, proposed to be the driver of carbon accumulation in Alaska, are less important, potentially explaining the apparently incongruous (high CAR) response to cold conditions during the LIA (Figs. 2, 3, 4). This is supported by the strong negative correlation between annual temperature range and CARs (Table 3). The strongest correlation is observed between CAR and isothermality, a measure of the difference between the annual range in temperature and the diurnal (daily) range in temperature (O'Donnell and Ignizio, 2012). In the Carpathians, a strong isothermality leads to high CARs, in line with measurements of carbon flux from peatlands, where strong isothermality has been linked to the shift of a bog to a net carbon sink (Webster et al., 2018). This may be related to the strong impact isothermality has on bryophyte distributions, with evidence to suggest range expansion of some species may occur under increasingly greater isothermality, especially in humid regions (Kou et al., 2020).

To investigate the impact of long-term hydroclimate variability on CARs we separate the model outputs into five 1000-year sections, compared to the composite record CARs for the same period (Table 3). From this exercise the strongest agreements appear between hydroclimate and CAR in the period 5000 – 4000 yr BP, with statistically significant correlations between CARs and the seasonality of precipitation, mean temperature, and maximum temperature, respectively. The strongest correlation is with maximum temperature (Fig. 4), supporting previous evidence that in the pre-anthropogenic world the strongest control over peat growth in this region has been linked to temperature variability (Charman et al., 2015, 2013). Laboratory, and field-based experiments have shown this to be the case, with greater temperatures leading to growth of larger *Sphagnum* plants, and an increase in the amount of biomass produced (Asada et al., 2003; Breeuwer et al., 2008;





Gunnarsson, 2005). In Romania, temperature was listed as the most important climatic variable for peat growth, as it led to the development to *Sphagnum*-dominated high-biomass peat for the TM record, rather than sedge-dominated (Panait et al., 2017). Strong correlations also exist between PAR0, a measure of photosynthetically active solar radiation, and *Sphagnum* growth (Loisel et al., 2012). The correlation with precipitation seasonality is also in line with previous work, which has
suggested that the availability of moisture during the growing season is a key control on peat growth (Jassey and Signarbieux, 2019; Oke and Hager, 2017), with seasonality linked to the total peat carbon stock preserved (Kurnianto et al., 2015). In this case, we assume the increase in seasonality results in a rise in summer precipitation, rather than winter (Fig. 4, Table 3).

No further statistically significant correlations are observed during any of the three subsequent 1000-year periods (3950 –
3000, 2950 – 2000 and 1950 – 1000 yr BP), indicating that climate is not playing the major forcing role on peat growth through this time (Fig. 4, Table 3). Such a finding supports our earlier assertion that human impact has become a primary driver of peat growth in the Carpathians, with our model-data comparison suggesting the shift occurs after 4000 yr BP, further supporting estimates of enhanced human pressure on the mid to high mountain environments (Giosan et al., 2012; Schumacher et al., 2016), which led to widespread soil erosion (Longman et al., 2017a).

Between 950 yr BP and present, there are strong positive correlations between CAR and mean and minimum temperatures, denoting a primarily climatic control on peat accumulation (Fig. 4, Table 4). However, the correlation in this period is driven by the last 200 years, where recent warm temperatures reflect recent high CARs (Figs. 2-4). These recent high temperatures largely reflect anthropogenic activity and climate change associated with greenhouse gas emissions, and it might be asserted that higher CAR over this period is not linked with natural climatic fluctuations, but rather to an increase in human
emissions. It must also be noted that it is possible recent CAR increases are also related to the speed of peat decomposition. Recent peat has undergone less decomposition resulting in high levels of carbon preservation (Young et al., 2019).

### 4.1 Comparison with other potential driving forces

Some previous work also highlighted a range of other drivers behind peat accumulation rates, with mineral dust input listed as the primary driver of peat growth in a Swedish peatland (Kylander et al., 2018). We use the two existing records of dust
deposition in the Romanian Carpathians to investigate this connection (Longman et al., 2017b; Panait et al., 2019) in our studied region (Fig. 5). A weak positive correlation between TM dust flux and our composite CAR record may be representative of this linkage, but the MOH dust flux record appears to show little or no correlation with the composite CAR (Fig. 5, Table 4). This positive correlation is enhanced when the dust flux of Panait et al. (2019) is compared to the individual CAR of TM (Fig. 5), with a highly significant positive correlation observed (Table 4). For MOH a weaker, but
still statistically significant correlation is present (Table 4) between the dust flux of Longman et al. (2017b) and our CAR (Fig. 5). These findings indicate that dust may be a local control on carbon accumulation, but that each site must be treated individually, since dust may be local in source, with certain sources potentially much more nutrient-rich than others (Kylander et al., 2018; Longman et al., 2017b). In particular, dust from Saharan sources is likely to contain primarily quartz




and carbonate clasts, which will not contain large quantities of phosphorous or nitrogen (Varga et al., 2016). Another factor
in the control of CAR by dust deposition is peat type, with ombrotrophic bogs considered very nutrient poor, and generally
phosphorous and nitrogen limited (Damman, 1986; Ratcliffe et al., 2020; Wang and Moore, 2014). This may explain the lack
of strong correlation between our CAR composite and two regionally representative dust flux records (Table 4).

We showed that both dust input and variability in hydroclimate may have played a role in controlling CARs in the study
region, but we hypothesise that human impact may have become the major controlling factor after 3000 yr BP. To
investigate this, we compare the CAR composite record with several land use and human impact-related proxies from the
HYDE dataset (Fig. 6). We focus on the last 2000 years of the record due to data availability, as prior to 2000 yr BP, only a
single data point for every thousand years is available from the HYDE dataset. However, these data are sufficient to indicate
the strong positive correlations between population and land use, and CARs in central-eastern Europe (Table 4). This is
further supported by the comparison with land use as reconstructed by Kaplan et al. (2011), and the strong positive
correlation between the two datasets (Fig. 6, Table 4). To supplement this, we also compare to a synthesis of eastern
European biomass burning (Feurdean et al., 2020). In the studied period, it appears biomass burning was controlled primarily
by human-driven tree felling and burning, and so may be used to estimate levels of land exploitation (Feurdean et al., 2020).
There are strong positive correlations between CARs and biomass burning in the last 2000 years (Fig. 6, Table 4). This
finding suggests biomass burning and associated forest removal has a positive impact on central and eastern European
peatland CARs. As described earlier, increased human impact (logging, use of fire) lead to reduced forest cover (Feurdean et
al., 2020) and increased weathering (Longman et al., 2017a). Greater biomass burning suggests greater deforestation, and the
possibility of more soil entrainment. This links to our assertion of a dust-related control, as local dust sources may be linked
to local deforestation and periodic topsoil exposure (Longman et al., 2017b; Panait et al., 2019).

This data comparison exercise clearly indicates the complexity of factors controlling mountain peat CARs, and prediction of
the behaviour of the peatland carbon sink is a challenge. Considering climate indices, it appears there is a relationship
between mean temperature and CAR, and potentially in the availability of moisture during the growing season (Table 4). In
Romania, average temperatures are predicted to increase in the next decades, with concurrent increases in summer and spring
precipitation, and decreases in winter (Ministry of Environment and Climate Change, 2013). As such, it is likely that these
conditions will lead to increased peat growth, and enhanced CARs. However, as we have shown, the primary control on
CARs for much of the last 4000 years has been the combination of human impact, local soil erosion and periodic deposition
of long-range transported dust. Changes in erosion are reliant upon a range of local factors, and dust fluxes are often
controlled by distal droughts, so it is not possible to predict how erosion rates will change in the future for each individual
bog location. Our work clearly demonstrates that CARs are not purely controlled by climate and indicates that predictions of
the future size of the global peatland carbon sink must be adjusted to take into account local factors relating to human
activities (Gallego-Sala et al., 2018).



## 5 Conclusions

Using nine peatbog records from the Carpathians, we compiled a detailed record of changing carbon accumulation rates (CARs) through the Holocene. Our data indicates that CARs for bogs located in the Carpathian Mountains are potentially very efficient carbon sinks, with the long-term apparent carbon accumulation rates amongst the highest in the published
literature, and comparable to high mountain bogs in South America. As a result, the peatland carbon sink in the Carpathians and throughout central-eastern Europe may be large, despite the small area of peatland coverage. Our work suggests that mountain bogs should be considered in future estimates of the peatland carbon sink, and that further studies are needed to refine estimates of carbon accumulation in such environments.

From our reconstructions, CARs varied throughout the Holocene, with high rates recorded in the early Holocene (10000 –
7000 yr BP), before a slow decline culminating in CAR nadir around 4000 yr BP. These results may be linked to the warm climate of the early Holocene stimulating peat growth before gradual cooling and the possible impact of the cold 4.2 kyr event. This is supported by comparison to palaeoclimatic model data from the period 5000 – 4000 yr BP, with strong correlations between temperature and CAR. After 3000 yr BP, there is a clear shift in controls on peat accumulation, with no apparent correlations between CARs and temperatures. Instead, we suggest late Holocene CARs are controlled by a
combination of human activity and dust input. A clear peak in CARs occurs at the same time as the Roman Warm Period and the rise of the Roman Empire, who would have introduced greater land use, and resulted in increased erosion and mineral input into bogs, supplying nutrient and resulting in increased CARs. Alongside this, increases in dust flux correlate well with individual CAR reconstructions, indicating changing dust sources and levels of dust to the region may have led to local CAR responses. In the most recent part of the records, sharp rises in CARs are reconstructed, potentially indicating the impact
warmer temperatures because of greenhouse gas emissions are having on peat growth.

### Data Availability

All data generated in the course of this study will be uploaded to Pangaea, and a doi will be added here once produced.

### Author contributions

Conceptualization, JL, DV, AH, VE, WF; Methodology, JL, DV, WF; Formal analysis, JL, DV, VE, AH, WF, DP;
Investigation, JL, DV, AH, WF, TS, DP, RB; Resources, DV, VE, TS, DP, RB; Writing – Original Draft, JL, DV, WF; Writing – Review & Editing, JL, DV, AH, VE, WF, TS; Visualisation, JL, AH; Funding acquisition, DV.

### Competing Interests

The authors declare that they have no conflict of interest.



**Acknowledgements**

We are indebted to S.B. Marković, N. Tomić, I. Obreht and Sz. Kelemen for fieldwork support. This work was supported by a grant of the Romanian Ministry of Education and Research, CNCS - UEFISCDI, project number PN-III-P4-ID-PCE-2020-0914, within PNCDI III.

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





**Figures**

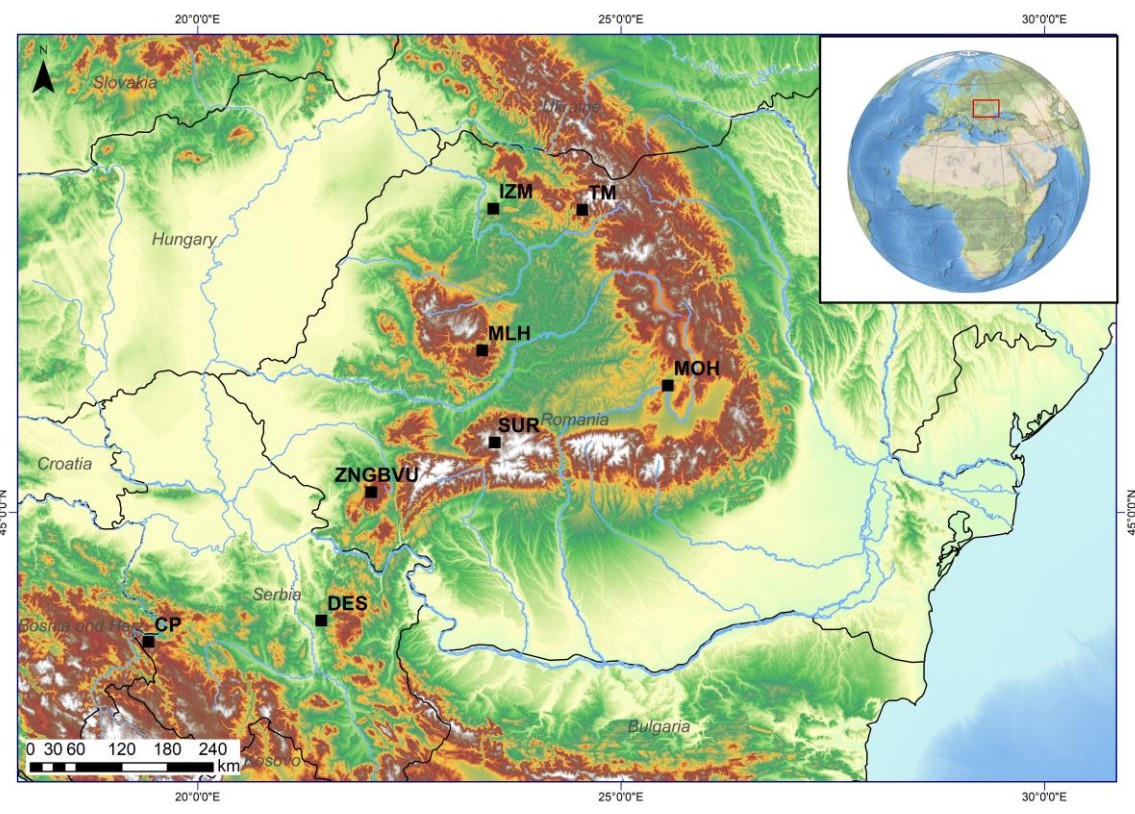

**Figure 1: Map of central-eastern Europe, indicating where all the sites used in this study are located. Study sites are marked by filled squares, with site codes indicating the site name: Iezerul Mare (IZM), Tăul Muced (TM), Mohos (MOH), Mluha (MLH), Sureanu (SUR), Zănoaga Roşie (ZNG), Baia Vulturilor (BVU), Despotovac (DES) and Crveni Potok (CP).**

735



Figure 2: Changing carbon accumulation rates (CARs) through time for all sites in this study, alongside the results of changepoint
modelling. Each panel denotes a single peat bog CAR record, moving south to north; A) Crveni Potok (CP), B) Despotovac (DES),
C) Zănoaga Roșie (ZNG), D) Baia Vulturilor (BVU), E) Sureanu (SUR), F) Mluha (MLH), G) Mohos (MOH), H) Tăul Muced
(TM) and I) Iezerul Mare (IZM). In each panel, individual measurements are denoted by filled circles. Changepoint model outputs
are shown by the solid coloured line indicating the mean and the 5% and 95% percentiles. Horizontal dashed lines indicate the
long-term carbon accumulation (LORCA) values. In each panel, the black solid line denotes the likelihood of changepoint
occurrence.





**Figure 3: Collated normalised carbon accumulation rates, and composite of data. Panel A displays the 50-year binned, z-score converted raw data from each site. Panel B shows the synthesis of all data in panel A, with the mean z-score indicated with a solid black line. 1 standard deviation is indicated with solid green lines. Panel C is changepoint modelling of the composite time series. As before, filled circles indicate datapoints. Changepoint model outputs are shown by the solid purple line indicating the mean and dashed purple lines indicating the 5% and 95% percentiles. The black solid line denotes the likelihood of changepoint occurrence.**





**Figure 4: Comparison of carbon accumulation rate (CAR) composite with climatic variables as extracted from our modelling**
**approach. In panels A-D are the variables of climate used in this study to compare with the reconstructed CAR for central-eastern**
**Europe (panel E). Highlighted with coloured rectangles are 1000-year periods in which a statistically significant correlation**
**between CAR and the variable of that colour are observed (e.g. precipitation seasonality and CAR in the period 1950 – 1000 yr BP,**
**in the panel D). When two variables significantly correlate with CARs, both colours are indicated by rectangles (e.g., mean and**
**minimum temperature, and CAR in the period 1000 yr BP to present in panel C). Also indicated and denoted by dashed lines are**
**the Roman Warm Period (RWP), Medieval Warm Period (MWP) and Little Ice Age (LIA).**

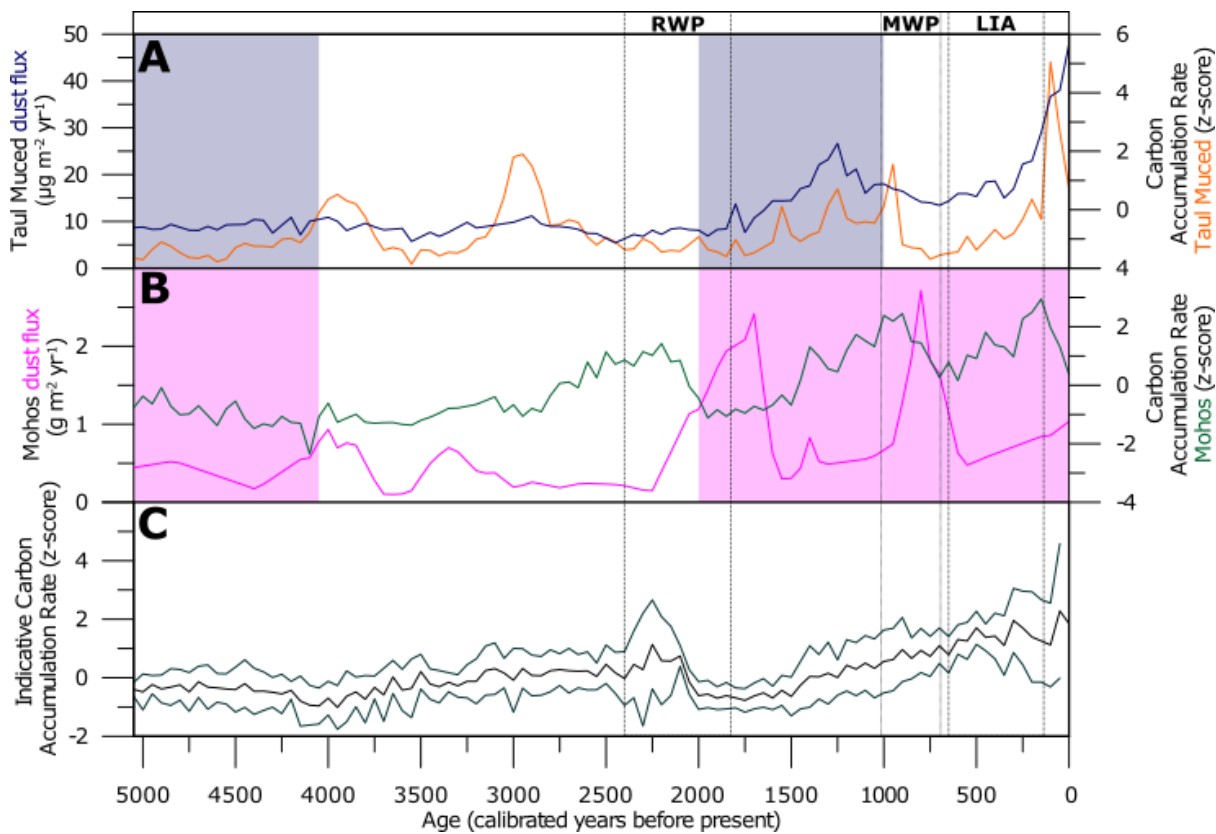

**Figure 5: Comparison of dust fluxes and our reconstruction of carbon accumulation rate (CAR). Panel A shows the dust flux**
**reconstructed from Tăul Muced, alongside the corresponding CAR. Panel B shows the dust flux reconstructed from Mohos,**
**alongside the corresponding CAR. Panel C shows the CAR composite record. Coloured rectangles indicate significant correlation**
**between the variable indicated by the colour and the CAR reconstruction, as in Figure 4. Also indicated and denoted by dashed**
**lines are the Roman Warm Period (RWP), Medieval Warm Period (MWP) and Little Ice Age (LIA).**





**Figure 6: Comparison of reconstructed carbon accumulation rate (CAR) for central-eastern Europe with four variables associated with land use and human impact. Panels A-E display the land use variables, with charcoal deposition in panel A (Feurdean et al., 2020), an estimate of total land use in panel B (from Kaplan et al., 2011), and three metrics of human impact (area of pasture, cropland and total population) in panels C-E (Goldewijk et al., 2017). Panel F displays the CAR reconstruction from this study. Coloured rectangles indicate correlation between the variable indicated by the colour and the CAR reconstruction, as in Figure 4.**



**Tables**

Table 1: Main results of this study. Listed here are the details of each bog involved in this work, their location, size, depth and age. Also presented are the average carbon content, bulk density and peat growth.

| Peat Bog | Code | Longitude | Latitude | Altitude | Mire area (km²) | Median C content (%) | Median density (g cm⁻³) | Basal Age (yr BP) | Basal Depth (cm) | Peat Growth (cm yr⁻¹) |
|---|---|---|---|---|---|---|---|---|---|---|
| Mohos | MOH | 46.05 | 25.55 | 1050 | 0.8 | 53.13 | 0.071 | 10849 | 940 | 0.086 |
| Sureanu | SUR | 45.581 | 23.507 | 1840 | 0.009 | 49.02 | 0.095 | 7390 | 594 | 0.080 |
| Mluha | MLH | 46.336 | 23.359 | 1240 | 0.1 | 57.03 | 0.204 | 7863 | 620 | 0.078 |
| Zanoaga Rosie | ZNG | 45.17 | 22.05 | 1400 | 1 | 57.65 | 0.18 | 11936 | 200 | 0.016 |
| Baia Vulturilor | BVU | 45.17 | 22.05 | 1400 | 1 | 53.29 | 0.11 | 4030 | 193 | 0.047 |
| Crveni Potok | CP | 43.91 | 19.42 | 1090 | 0.19 | 52.25 | 0.14 | 9560 | 270 | 0.028 |
| Taul Muced | TM | 47.47 | 24.54 | 1360 | 0.02 | 56.92 | 0.074 | 7769 | 450 | 0.057 |
| Iezerul Mare | IZM | 47.48 | 23.49 | 1005 | 0.1 | 52.95 | 0.18 | 6177 | 750 | 0.121 |
| Despotovac | DES | 44.09 | 21.46 | 100 | 0.001 | 20.32 | 0.35 | 1779 | 200 | 0.112 |

Table 2: Average carbon accumulation rates for a selection of times periods. Also displayed are the LORCA value and an estimate of carbon storage. See text for details.

| Peat Bog | Code | RERCA (50 yr BP to Present) | Gallego-Sala approach (1000 - 100 yr BP) | Medieval Warm Period (1000-700 yr BP) | Little Ice Age (650-100 yr BP) | Roman Warm Period (2200-1550 yr BP) | LORCA (g C m-2 yr-1) | Estimate of C storage (Gt) |
|---|---|---|---|---|---|---|---|---|
| Mohos | MOH | 39.29 | 42.18 | 46.45 | 40.88 | 25.32 | 32.61 | 0.00028 |
| Sureanu | SUR | 51.59 | 40.80 | 49.60 | 34.61 | 58.50 | 37.45 | 0.00000 |
| Mluha | MLH | 228.56 | 217.67 | 191.24 | 235.78 | 74.03 | 91.74 | 0.00007 |
| Zanoaga Rosie | ZNG | 159.61 | 27.06 | 24.99 | 28.58 | 17.36 | 17.12 | 0.00021 |
| Baia Vulturilor | BVU | 26.67 | 29.34 | 20.44 | 33.19 | 21.79 | 28.08 | 0.00011 |
| Crveni Potok | CP | | 29.89 | 41.19 | 20.85 | 47.77 | 20.99 | 0.00004 |
| Taul Muced | TM | 96.45 | 37.02 | 33.99 | 39.28 | 19.22 | 24.32 | 0.00000 |
| Iezerul Mare | IZM | 283.45 | 217.47 | 224.64 | 211.18 | 115.57 | 120.59 | 0.00007 |
| Despotovac | DES | 15.39 | 74.18 | 35.46 | 96.94 | 7.57 | 81.11 | 0.00000 |

Table 3: Correlation coefficients and p-values between bioclimatic parameters of climate change and reconstructed carbon accumulation rates, for a selection of times periods. P-values significant at the 0.05 level are highlighted in bold, with those significant at the 0.01 level italicised.

| Whole period (5000 yr BP to present) | Pearson's r | p-value |
|---|---|---|
| Mean Precipitation | 0.017 | 0.87 |





| | | |
|---|---|---|
| Precipitation Seasonality | -0.326 | *0.00* |
| Mean Temperature | 0.141 | 0.16 |
| Minimum Temperature | -0.073 | 0.46 |
| Maximum Temperature | 0.371 | *<0.01* |
| Diurnal range | 0.339 | *<0.01* |
| Annual range | -0.421 | *<0.01* |
| Isothermality | 0.505 | *<0.01* |
| Temperature seasonality | -0.382 | *<0.01* |
| **5000-4000 yr BP** | **Pearson's r** | **p-value** |
| Mean Precipitation | 0.304 | 0.17 |
| Precipitation Seasonality | 0.477 | **0.03** |
| Mean Temperature | 0.52 | *0.01* |
| Minimum Temperature | 0.263 | 0.24 |
| Maximum Temperature | 0.547 | *0.01* |
| Diurnal range | 0.169 | 0.45 |
| Annual range | 0.251 | 0.26 |
| Isothermality | -0.183 | 0.42 |
| Temperature seasonality | 0.307 | 0.17 |
| **3950-3000 yr BP** | **Pearson's r** | **p-value** |
| Mean Precipitation | -0.119 | 0.60 |
| Precipitation Seasonality | -0.397 | **0.07** |
| Mean Temperature | -0.007 | 0.97 |
| Minimum Temperature | 0.159 | 0.48 |
| Maximum Temperature | -0.179 | 0.43 |
| Diurnal range | -0.326 | 0.14 |
| Annual range | -0.259 | 0.25 |
| Isothermality | 0.366 | 0.10 |
| Temperature seasonality | -0.35 | 0.11 |
| **2950-2000 yr BP** | **Pearsons' r?** | **p-value** |
| Mean Precipitation | -0.168 | 0.46 |
| Precipitation Seasonality | 0.038 | 0.87 |
| Mean Temperature | -0.025 | 0.91 |
| Minimum Temperature | -0.095 | 0.68 |
| Maximum Temperature | 0.264 | 0.24 |
| Diurnal range | 0.283 | 0.20 |
| Annual range | -0.034 | 0.88 |
| Isothermality | 0.101 | 0.66 |
| Temperature seasonality | -0.022 | 0.92 |
| **1950-1000 yr BP** | **Pearson's r** | **p-value** |
| Mean Precipitation | -0.169 | 0.45 |





| | Pearson's r | p-value |
|---|---|---|
| Precipitation Seasonality | -0.365 | **0.10** |
| Mean Temperature | -0.258 | 0.25 |
| Minimum Temperature | -0.185 | 0.41 |
| Maximum Temperature | 0.171 | 0.45 |
| Diurnal range | 0.355 | 0.11 |
| Annual range | -0.061 | 0.79 |
| Isothermality | 0.203 | 0.37 |
| Temperature seasonality | 0.143 | 0.53 |
| **950 yr BP to present** | **Pearson's r** | **p-value** |
| Mean Precipitation | -0.206 | 0.36 |
| Precipitation Seasonality | 0.103 | 0.65 |
| Mean Temperature | 0.465 | **0.03** |
| Minimum Temperature | 0.497 | **0.02** |
| Maximum Temperature | 0.12 | 0.60 |
| Diurnal range | -0.533 | *0.01* |
| Annual range | -0.295 | 0.18 |
| Isothermality | -0.016 | 0.94 |
| Temperature seasonality | -0.184 | 0.41 |



**Table 4: Correlation coefficients and p-values between indicators of human impact, and of dust input and reconstructed carbon**
785 **accumulation rates, for a selection of times periods. [a] Correlation of dust flux from Mohos and Tăul Muced bogs with composite**
**CAR record. [b] Correlation of dust flux from Mohos and Tăul Muced with individual CARs. P-values significant at the 0.05 level**
**are highlighted in bold, with those significant at the 0.01 level italicised.**

| Whole period (5000 yr BP to present) | Pearson's r | p-value |
|---|---|---|
| HYDE Population | 0.584 | *<0.01* |
| HYDE Cropland | 0.652 | *<0.01* |
| HYDE Pasture | 0.881 | *<0.01* |
| Kaplan land use | 0.673 | *<0.01* |
| MOH Dust Flux and CAR composite[a] | 0.042 | 0.67 |
| TM Dust Flux and CAR composite[a] | 0.240 | **0.02** |
| MOH Dust and MOH CAR[b] | 0.220 | **0.03** |
| TM Dust and TM CAR[b] | 0.598 | *<0.01* |
| Charcoal and composite CAR[c] | 0.774 | *<0.01* |

| 5000-4000 yr BP | Pearson's r | p-value |
|---|---|---|
| HYDE Population | n/a | n/a |
| HYDE Cropland | n/a | n/a |
| HYDE Pasture | n/a | n/a |
| Kaplan land use | n/a | n/a |
| MOH Dust Flux and CAR composite[a] | -0.518 | *0.01* |
| TM Dust Flux and CAR composite[a] | -0.691 | *<0.01* |
| MOH Dust and MOH CAR[b] | 0.052 | 0.82 |
| TM Dust and TM CAR[b] | 0.398 | 0.07 |
| Charcoal and composite CAR[c] | n/a | n/a |

| 3950-3000 yr BP | Pearson's r | p-value |
|---|---|---|
| HYDE Population | n/a | n/a |
| HYDE Cropland | n/a | n/a |
| HYDE Pasture | n/a | n/a |
| Kaplan land use | 0.826 | *<0.01* |
| MOH Dust Flux and CAR composite[a] | -0.418 | **0.05** |
| TM Dust Flux and CAR composite[a] | -0.216 | 0.33 |
| MOH Dust and MOH CAR[b] | 0.472 | 0.03 |
| TM Dust and TM CAR[b] | 0.684 | *<0.01* |
| Charcoal and composite CAR[c] | -0.733 | *<0.01* |



| 2950-2000 yr BP | Pearson's r | p-value |
|---|---|---|
| HYDE Population | n/a | n/a |
| HYDE Cropland | n/a | n/a |
| HYDE Pasture | n/a | n/a |
| Kaplan land use | -0.218 | 0.33 |
| MOH Dust Flux and CAR composite[a] | -0.632 | *<0.01* |
| TM Dust Flux and CAR composite[a] | 0.658 | *<0.01* |
| MOH Dust and MOH CAR[b] | -0.057 | 0.80 |
| TM Dust and TM CAR[b] | 0.760 | *<0.01* |
| Charcoal and composite CAR[c] | 0.369 | **0.09** |

| 1950-1000 yr BP | Pearson's r | p-value |
|---|---|---|
| HYDE Population | 0.480 | 0.02 |
| HYDE Cropland | 0.019 | 0.93 |
| HYDE Pasture | 0.731 | *<0.01* |
| Kaplan land use | -0.470 | **0.03** |
| MOH Dust Flux and CAR composite[a] | -0.460 | **0.03** |
| TM Dust Flux and CAR composite[a] | 0.243 | 0.28 |
| MOH Dust and MOH CAR[b] | -0.538 | *0.01* |
| TM Dust and TM CAR[b] | 0.807 | *<0.01* |
| Charcoal and composite CAR[c] | 0.948 | *<0.01* |

| 950 yr BP to present | Pearson's r | p-value |
|---|---|---|
| HYDE Population | 0.571 | *0.01* |
| HYDE Cropland | 0.569 | *0.01* |
| HYDE Pasture | 0.465 | **0.03** |
| Kaplan land use | 0.598 | *<0.01* |
| MOH Dust Flux and CAR composite[a] | 0.497 | **0.02** |
| TM Dust Flux and CAR composite[a] | 0.120 | 0.60 |
| MOH Dust and MOH CAR[b] | -0.142 | 0.53 |
| TM Dust and TM CAR[b] | 0.717 | *<0.01* |
| Charcoal and composite CAR[c] | 0.721 | *<0.01* |