# Peer review of "Carbon accumulation rates of Holocene peatlands in central-eastern Europe document the driving role of human impact for the past 4000 years"

_Climate of the Past, 2021_

## Author Response (AR1)

In this manuscript Longman et al., present eight new carbon accumulation records for peatlands in the Carpathian mountains and combine this with the one existing published record. Not only is this a highly novel paper due to the location and type of peatland analysed it is also an excellent contribution to the question of the drivers of long-term carbon accumulation in peatlands. Longman et al., have done an admirable job of untangling the complex drivers of peatland carbon accumulation both in their own data and in the earlier literature. The difference in dominant drivers between the early and mid-to-late Holocene is highly convincing and matches similar trends seen elsewhere for which others have thus far failed to find such a satisfactory explanation.

*We appreciate the reviewer's comments, and have attempted to respond to all all remarks and suggestions in the document below. Our responses are highlighted in italics.*

I find the main message of this paper quite convincing, that message being that long term peatland C can be controlled by either climate or nutrient factors and that indeed a combination of these two drivers are important. Although I am myself convinced of the direct role dust and nutrients can have on CAR, if I am to be the 'devils advocate' the authors may wish to consider and rule out the linkage between dust and/or mineral input and climate itself which presents a conundrum as to which is the direct or dominant driver of CAR. Dust inputs into peatlands have in themselves been used to reconstruct climate, with the expectation that increasing aridity results in more dust (e.g. https://doi.org/10.1016/j.epsl.2009.03.013). This raises the question about how independent dust inputs and climate really are. In the case of my own work in New Zealand it was quite easy to discount climate warming or aridity as we would not expect this after a volcanic eruption, but for this work the separation is less clear. This dataset might offer a simple opportunity to test this given the detailed climate and dust records the authors present.  It might warrant inclusion of a new regression or correlation table in the supplementary material or at the least a sentence or two considering how dust and climate may be interlinked and how the authors can separate these as dominant drivers.

*This is a valuable suggestion, and we have attempted to discuss the point in the updated manuscript. We now compare the record of dust flux from the Mohos bog (selected due to its length) to all the reconstructed variables from Paleoview. These comparisons (in the form of a list of regression values) are now included in the supplementary information.*

*This exercise clearly shows the linkage between climate and dust deposition generally, and specifically in the earliest section of the compared record (4950 – 4000 yr BP). However, the disconnection between the two in the most recent 2000 years is also clear, suggesting the dust input is not purely climatically controlled, rather that it reflects a number of controls. These may include deforestation and human impact, thereby supporting our argument.*

*To reflect this point in the updated manuscript, we have added the following to the text:*

*"However, it has been proposed changing dust flux is purely related to climatic controls (e.g. Marx et al., 2009), and so To further investigate a potential controlling impact of local dust and erosion, we compare the model data to the reconstruction of dust flux from Mohos (Supplementary Table 3). This exercise clearly indicates the covariance of dust and climatic forcing factors (and especially precipitation) when considering the whole record. However, for the periods in which we infer local erosion and dust supply stimulating peat growth, there are either very weak, or not statistically significant correlations between climate and dust (Supplementary Table 3). Such a finding indicates the disconnection of climate and dust flux in the last 2000 years, and supports our assertion of local dust drivers in this period."*

*We also add the correlation table to the supplementary information (Supplementary Table 3).*

In addition to this point above I have some minor comments and suggestions, see below.

Figure 2. My personal preference here would be to reverse the x axis so the time goes from modern to older, but this is just a personal preference and should be the authors decision.

*We appreciate the author's comment but believe that working from left (oldest) to right (youngest) is more suitable, and so we will not be making this change.*

Supplementary Figures:

The inclusion of C content and bulk density records for the individual cores would be nice to see in the supplementary figures. Particularly how often and to what degree mineral material is being washed in and also, thinking for the future, this information could be useful for future synthesis papers regarding peat properties.

*All this data will be uploaded to Pangea as part of our updated submission and so will be available to download for any potentially interested readers.*

Methodology:

It should be stated somewhere that the AMS radiocarbon dates are primarily on bulk peat. This has been controversial in the past, however recently I think people are less concerned about this https://doi.org/10.1016/j.quageo.2015.10.003 , however the situation could still change so it would be good to be clear about this in the methodology, not just the SI.

*We have added a sentence to clarify this in the main text:*

*"The majority of samples were taken from bulk sediment/peat, an approach which has been shown to yield reliable age information (Holmquist et al., 2016)."*

L19: Suggest for clarity changing to "mountainous peatlands"

*We have made this change.*

L28: This is really interesting!

*We appreciate the reviewer's enthusiasm.*

L43: Suggest you say "stimulates plant growth more than organic matter decomposition"

*Change made.*

L52: This question of early human influence is extremely interesting if a little tangential, I remember a recent paper revising the ancient human population upwards considerably. If you have not seen it I quite like the following paper about the 'lost' medieval peatlands in Flanders Belgium.  https://doi.org/10.1007/s12685-011-0037-4

*We agree the question of when humans began to alter their environment is a very interesting one, and appreciate the paper recommendation.*

L62: I don't remember Kylander et al., 2018 talking about the inputs of mineral soil into peatlands due to deforestation and there is not any mention of trees or forests in that paper that I can find using ctrl+f. It might be worth checking that reference. It's a very interesting idea though!

*We have restructured this sentence to reflect the two points being discussed:*

*"Finally, it is possible that increased erosion as a result of forest removal either by human actions or natural causes may lead to increased mineral in-wash (Longman et al., 2017a), with dust deposition potentially stimulating peat growth (Kylander et al., 2018)."*

L72: Might be worth mentioning here how important these mires may be regionally for Romania/Serbia's carbon inventory, especially as so little carbon work has been done on them

*We have added the following to the sentence:*

*"...one which may represent a regionally important carbon stock.  "*

L194: Also maybe worth mentioning that these numbers are more comparable to those from Eddy Covariance. It's also really cool that we have these high numbers now for down-core and gas measurements of C accumulation

*We have added the following to this sentence:*

*"Further, eddy covariance data regularly indicates carbon accumulation rates greater than 100 g C m$^{-2}$ yr$^{-1}$ (e.g. Roulet et al., 2006)."*

L194: You might also want to consider this paper https://doi.org/10.1016/j.quascirev.2019.03.022 where there was also comparably high C accumulation despite a relatively harsh climate.

*We have cited the suggested paper here in the updated manuscript.*

L195: The nutrients are presumably coming in in 'pulses' which seems to be quite important according to a new pre-print https://doi.org/10.31223/X5FW3J

*We have added the suggested reference to this section and discuss the possibility of pulsing nutrient supply being important.*

*"In each potential scenario, nutrients would likely be supplied in pulses, shown to be important for peat growth (Schillereff et al., 2021)."*

L269: This could be a good place to mention the geology and soil fertility in the Carpathians. Is it exceptional in any way?

*We have added the following to the manuscript:*

*"In parts of the study area such as the eastern Carpathians, soils are dacitic and contain large quantities of volcanic minerals, meaning they are potentially nutrient-rich (Longman et al., 2017b)."*

L271: Please define the migration period, I am not familiar with this

*We have defined this in the updated manuscript.*

L308: This also contradicts the following highly cited paper: https://doi.org/10.1038/s41558-018-0271-1 however I can really believe this is the case given the bias towards regions with cold and continental climate

*We have added this reference at this point, and altered the following sentence to reflect the fact that this observation is not just from Alaska, but across high latitude regions.*

L346: I agree it is not possible to read too much into the recent changes for the reasons you have mentioned. I also recommend you remove the last sentence of the conclusion for this reason (L408)

*We have removed this line from the updated manuscript.*

**Response to Reviewer 2**

This study by Longman et al. present a thorough synthesis of long-term Holocene carbon accumulation rates for the Carpathian Mountains, presenting data from eight new peat cores. This is an excellent addition to global peat datasets as records from mountain peatlands and peatlands from Eastern Europe and poorly studied. The Holocene dating resolution for the study sites is very impressive, as is the evaluation of the potential drivers for changes in CAR. The methods and discussion of a complex mix of drivers including anthropogenic impacts and dust inputs are well thought out and clearly presented, both from data-driven hypotheses and explanations from other studies in the region.

*We appreciate the reviewer's detailed and helpful comments, and have used them to improve the manuscript. Our responses to the comments are highlighted below in italics.*

In general, for clarity, I would recommend that the site information, results, supplementary etc. are presented in the same site order every time, either from North to South or the reverse.

*We agree this improves the readability of our work and have adjusted all figures and tables to present sites along a SW to NE transect, in the same manner as Figure 2.*

In addition, after reading such a well-presented and convincing "carbon story", I would have preferred the conclusion to end with larger-scale impacts for the future of the regions and further study. Perhaps adding that high-resolution studies of the last millennium or last couple centuries would be useful future studies to evaluate these drivers as well as recent anthropogenic impacts and future trajectories. Finally, the importance of the sites should be highlighted. Yes, it is a small carbon sink on a global scale, but what about the relative % for Romania or the Carpathian Mountain region?

*We appreciate these suggestions, and now include a short addition to the conclusion, which highlight how important these environments may be for CE European carbon storage and some possible future research directions.*

*"Our work indicates the potentially large carbon sink represented by mountainous peatlands in Central-Eastern Europe. To better constrain their importance in the regional carbona cycle, more work is necessary to investigate other carbon sinks in Romania and Serbia, such as lake sediment, forest soils and lowland peatlands. Work in these environments should help to better understand the size of the current carbon sink, and how it has varied in the past. This would help in determining exactly how important mountain bogs are in the regional carbon balance, and, when combined management and monitoring exercises, may help to mitigate the worst of the impacts of anthropogenic climate change."*

*Unfortunately, the data necessary to assess exactly how much the carbon sink of bogs in Romania represents as a proportion of total carbon sinks, and so we cannot make a reliable estimate. Future work in the region could make such conclusions possible.*

In addition to the above general comments, I have the following minor points to clarify the methods and enhance the discussion, in particular relating to the age-depth models.

Line 19: suggest rephrasing to "in mountainous peatlands"

*We have made this change.*

Line 49: suggest rephrasing to "on individual scales" to "for individual peatlands" or "on a local scale"

*We agree and have made this change.*

Line 72: suggest rephrasing to "important carbon sink for the regional carbon budget"

*We have rephrased this sentence in light of the other reviewer's comments, so it now reads:*

*"...constitute an important carbon sink for the overall carbon budget, one which may represent a regionally important carbon stock."*

METHODOLOGY

Section 2.1 Refer to Figure 1 and Table 1 here in the text

*We have added references to the figure and table here.*

Section 2.2: where were the samples for the other cores dated? What sample thickness was dated?

*We have included this information in the updated manuscript:*

*"The majority of samples were taken from 1cm slices of bulk sediment/peat, an approach which has been shown to yield reliable age information (Holmquist et al., 2016)."*

*And*

*"Samples from Sureanu (SUR), Zanoaga Rosie (ZNG), MLH and Mohos (MOH) bogs were measured at the HEKAL AMS Laboratory, MTA ATOMKI Institute for Nuclear Research of the Hungarian Academy of Sciences in Debrecen. 5 dates from SUR were analysed by the 14C Centre at Queen's University Belfast (Longman et al., 2017a). Samples from Crveni Potok (CP) were analysed on extracted plant macrofossils at the Poznan Radiocarbon Laboratory (Finsinger et al., 2017)."*

Section 2.3: There is a step missing here. To clarify, as with LOI calculations, was the C density calculated as 50% of the organic matter density (as in Turunen et al. 2002 and others by convention)?

*We did indeed follow this methodology, and include the following in the updated manuscript:*

*"Organic content was converted to carbon density by assuming 50% carbon (c.f. Turunen et al., 2002)."*

It might be useful to present figures for the bulk density, C%, ash content for each core in the supplementary material, particularly in light of the focus in the discussion on dust.

*We include these figures in the updated supplementary material.*

Section 2.4: The time series with changepoint analysis aspect is really interesting!

I suggest that the LORCA and C stock sections be moved to section 2.3 (on Carbon accumulation calculations). There should also be a mention that these rates (CAR,

LORCA) are all "apparent" rates, not accounting for decomposition, and not calculated from a net carbon balance.

*We have moved this methodology to Section 2.3, and now indicate these rates are all 'apparent':*

*"As we do not account for decomposition, all rates presented here (e.g. LORCA, RERCA and CAR) are apparent and not calculated from a net carbon balance."*

Line 104: Clarify the simple LORCA definition before explaining the calculation. As in the Clymo and Turunen articles cited, LORCA is the cumulative carbon for the core divided by the basal age.

*We clarify this in the updated manuscript.*

Line 117: The justification for limiting discussion of RERCA is very relevant here – focus on longer term rates! I would suggest adding a methodological justification that the dating resolution for the last ~100 years is limited for the cores presented.

*We have updated the manuscript as follows:*

*"Further, the resolution of radiocarbon dating for the most recent 100 years is too low for meaningful conclusions for a number of the bogs. Therefore, we also quantified the amount of carbon accumulation for the past 1000 years (see Table 2 for all average values), following Gallego-Sala et al. (2018), and focus our discussion on longer-term CAR variability."*

Line 135: "averaging multiple records provides" (add "s" to provide)

*Change made.*

RESULTS

I suggest, for clarity, that the results in the figures, models and tables be presented in the same order as in the methods.

*The ordering of the methods is dependent upon location of analyses (e.g. where the radiocarbon and LOI analysis was completed), and so we would rather not use this ordering scheme throughout. Instead, and in agreement with the reviewer's earlier comment, we now order all figures, model and tables from SW to NE.*

Line 170: Suggest using consistent decimals for LORCA and CAR.

*We have removed all decimal places here, for simplicity and uniformity.*

Line 186: peat bogs (separate the words)

*Change made.*

DISCUSSION

In addition to mineral inputs from local erosion and deposition, could wind direction and/or exposure be factors to consider? For instance, in peatlands along the Gulf of St Lawrence in Canada had higher LORCA if they were sheltered from cold winter winds (https://doi.org/10.1177/0959683614540727), and other peatlands in NE China had increased productivity from mineral inputs from wind-borne dust from the Loess Plateau (https://doi.org/10.1177/0959683619892661). Note that, while I do find the anthropogenic driver argument provided by the authors very convincing, these could be additional climatic factors to consider.

*These are interesting points, and we now refer the reader to the first study in our updated manuscript, but make it clear that the investigation of these small-scale forces is challenging in a study like ours:*

*"In addition to general climatic controls, it is also possible that smaller-term fluctuations and local configurations may impact CARs. For example, CARs of some peatlands along the St. Lawrence River in Canada are strongly impacted by the level of sheltering from cold winter winds (Magnan & Garneau, 2014). Such local climatic controls are challenging to evaluate on the scales studied here but may play a role in some of the bogs."*

*We have added the second reference to our section discussing the impact of dust-related nutrient supply:*

*"Some previous work also highlighted a range of other drivers behind peat accumulation rates, with mineral dust input listed as the primary driver of peat growth in a Swedish peatland (Kylander et al., 2018), and with fluctuations in supply of dust from the Loess Plateau in China important in controlling local peat development (Pratte et al., 2019)."*

Line 210: yes, this is a small global sink and the rapid nature of accumulation is great but perhaps to further value the importance of this sink, could you put it in context in Romania? For ex. Is this more C than in forests? Is it 50% of annual emissions?

*This is a good suggestion and so we have added the following to the updated manuscript. We have also changed the unit to megatonnes as it is more sensible for our values:*

*"It should be noted that this upper estimate is four times the amount of carbon sequestered by forests in Romania per year (2.5 Mt C/yr) (Olofsson et al., 2009), and represents 13% of the yearly Romanian anthropogenic carbon emissions (circa 80 Mt C/yr) (Crippa et al., 2021)."*

Line 345-346: Rephrase this passage for clarity. The decomposition is ongoing and recent peat has undergone less decomposition resulting in more "apparent" carbon. It's not clear that it is being preserved or sequestered.

*We have adjusted this sentence as suggested.*

TABLES

Table 1. I suggest (for clarity) that the authors present the peatlands in Table 1 in the same order as in the map or figures throughout the paper.

*We have made this change to Tables 1 & 2, and Figure 3.*

It may also be useful to add the coring date and a very general description of peatland type (raised bog, blanket bog, fen) – perhaps split into 2 tables (for example, Table 1: site information; Table 2: peat data), or in the supplementary material if the authors feel this is too much for one in-text table.

*We have added this information to Table 1, wherever it was available.*

Table 2. Add the units for RERCA

*Units added*

Table 3. Precipitation seasonality p-value for whole period should be <0.01

*Correction made.*

SUPPLEMENTARY MATERIAL

First paragraph: Rephrase - Sphagnum stems not stalks

*Change made.*

Last paragraph: The dates were calibrated using IntCal20 (Calib 8.2?) not IntCal13, as stated in the main text. Indicate here also that the models were generated using the rbacon package (v.?) in R.

*Changes made.*

Age-depth models: As stated previously, the modelling resolution is very impressive! Below are some small points of clarification, or curiosity on my part.

- Supplementary Figures 1, 4 and 3: indicate in the legends that the bottom of these models is interpolated (e.g. BVU, the bottom-most 14C date is 144 cm yet the model extends to almost 200 cm and the basal peat). The rates calculated for these interpolated sections could bias the results.

*We have added the following to each of these legends:*

*"The base of this model (between the last radiocarbon date and basal peat) is interpolated, and so carbon accumulation rates calculated from these depths should be treated with caution."*

- Supplementary Figures 6, 7 and 8: there is no surface input into these models. Were the dates interpolated from the top-most 14C date to the coring date? A brief sentence explaining this could be added to the legend.

*We have added the following to the legends of Figures 6 & 7:*

*"For the uppermost section of peat, the model interpolates between coring date and the first radiocarbon date."*

*For Supplementary Figure 8, sampling did not include the very uppermost section of peat, with the top date taken from the uppermost sampled peat layer.*

- Supplementary Figure 2 (SUR): what is the green calendar date at ~35-40 cm?

*This is the first radiocarbon date, coloured green by Bacon as it is younger than 1950 CE.*

- Supplementary Figure 6 (ZNG): Do the authors have any thoughts on why the accumulation rate is so low between 1000-8000 cal BP? Perhaps a hiatus in the model would allow for all the dates to be included, if there were a known disturbance such as erosion or a fire.

*This is not necessarily a hiatus, but a period of very slow peat growth, a feature observed in all blanket bogs from Semenic Mountain (with BVU showing it in deeper sections, but not presented here). A complete hiatus does not occur- indeed our radiocarbon dating across the period indicates the slow but steady accumulation of peat.*

Supplementary Table 1:

- It would be useful to add columns for calibrated dates and modelled dates (and ranges) in addition to the 14C ages presented in the table. If not here in the supplementary material then definitely on Pangaea

*We agree and include these data in full in our Figshare files.*

- There is an error with the depth for the Mlhua core – basal depth is 6730 cm!

*Change made- 'depth' and 'age' columns have been switched.*

Supplementary Figure 9: Romanian peat bogs (add n to Romanian)

*Change made.*